# Imaging Recommendations for Diagnosis, Staging, and Management of Central Nervous System Neoplasms in Adults: CNS Metastases

**DOI:** 10.3390/cancers16152667

**Published:** 2024-07-26

**Authors:** Kajari Bhattacharya, Abhishek Mahajan, Soujanya Mynalli

**Affiliations:** 1Department of Radiodiagnosis, Tata Memorial Hospital, Parel, Mumbai 400012, India; kajaribhattacharya7@gmail.com (K.B.); soju.mynalli@gmail.com (S.M.); 2Department of Imaging, The Clatterbridge Cancer Centre NHS Foundation Trust, 65 Pembroke Place, Liverpool L7 8YA, UK; 3Faculty of Health and Life Sciences, University of Liverpool, Liverpool L69 3BX, UK

**Keywords:** CNS metastases, MR imaging, adult brain metastasis, artificial intelligence

## Abstract

**Simple Summary:**

Brain tumors that spread from cancer elsewhere in the body (brain metastases) are becoming more common due to better cancer detection. While historically the outlook for patients with brain metastases was poor, new minimally invasive surgeries and targeted therapies are improving prognoses. Imaging features also depict responses to treatment and help identify the cause of patients’ symptoms worsening due to disease or therapy. The development of new MRI techniques, the use of artificial intelligence, and advanced therapeutic delivery systems are creating even more powerful tools that are directly contributing to improving patient care and better survival for patients with brain metastases. This article explores the screening and diagnostic and prognostic roles of various imaging modalities, and recommends standard guidelines in the management of cancer patients to improve the overall survival rate.

**Abstract:**

Brain metastases (BMs) are the most common central nervous system (CNS) neoplasms, with an increasing incidence that is due in part to an overall increase in primary cancers, improved neuroimaging modalities leading to increased detection, better systemic therapies, and longer patient survival. Objective: To identify cancer patients at a higher risk of developing CNS metastases and to evaluate associated prognostic factors. Methods: Review of imaging referral guidelines, response criteria, interval imaging assessment, modality of choice, as well as the association of clinical, serological, and imaging findings as per various cancer societies. Results: Quantitative response assessment of target and non-target brain metastases as well as an interval imaging protocol set up based on primary histological diagnosis and therapy status are discussed as per various cancer societies and imaging programs. Conclusion: Predictive factors in the primary tumor as well as independent variables of brain metastases like size, number, and response to therapy are necessary in management. The location of CNS metastases, symptomatic disease, as well as follow up imaging findings form a skeletal plan to prognosticate the disease, keeping in mind all the available new advanced therapy options of surgery, radiation, and immunotherapy that improve patient outcome significantly.

## 1. Introduction

Brain metastases (BMs) are the most common central nervous system (CNS) neoplasms, with an increasing incidence that is due in part to an overall increase in primary cancers, improved neuroimaging modalities leading to increased detection, better systemic therapies, and longer patient survival. The presentation of BM is varied and the diagnosis has always been sobering, with a guarded prognosis and management consisting of multimodality therapy including radiation and resection. In this era of minimally invasive surgery and targeted molecular immunotherapy, the prognosis seems to be improving on the basis of multiple variables, mostly determined on imaging and histology [1]. Here, we intend to acquaint the reader with imaging recommendations for CNS secondaries.

## 2. Risk Factors, Epidemiology

The incidence of BMs is about 150,000–200,000/year, 3–10 times that of primary malignant brain tumors [2]. BMs are seen in 10–30% of all cancer patients, commonly in lung (40–50%), breast (15–20%), and melanoma (5–20%), with the highest propensity in melanoma (50%) [3]. They are associated with a poor overall survival (OS) of about 8% at 2 and 2.5% at 5 years after diagnosis, the worst with leptomeningeal disease (LMD), OS < 6 months [4]. Young females with advanced adenocarcinoma of the lung, Her2+ve breast cancer at >41 years of age, elderly males with nodular desmoplastic/spindle cell melanoma in head–trunk, and advanced clear cell renal carcinoma are some established risk factors for developing BMs [5]. With molecular and genomic studies, it is evident that EGFR, Her2, BRAF, and ALK mutations are associated with BMs, which paves the way for targeted therapy [6].

## 3. Clinical Presentation

A significant proportion of these lesions are silent and hence are incidentally detected or are picked up only on screening studies. Presenting symptoms include headache (60%), focal neurologic deficit, seizure, cognitive impairment, or stroke-like symptoms, depending on number, size, and location. In contrast to lung cancer where BMs are detected synchronously or early, in breast cancer they arise in later period. Due to hematogenous spread and acquired mutations in primary to cross the blood–brain barrier (BBB), 80% occur in the cerebrum, 15% in the cerebellum, and 5% in the brainstem [7].

## 4. Clinical Work Up

In 15% of cases, the primary site is unknown [8]. Thus, a thorough physical examination (including testes and skin inspection), Computed tomography (CT) of the chest/abdomen, and sono-mammography are recommended and, if negative, whole-body PET is performed. The introduction of whole-body diffusion weighted imaging (WB-DWI) MRI with background body signal suppression has become feasible to detect primary and metastatic malignancies with high contrast resolution. However, exact localization of lesions may be less accurate due to a lack of anatomical reference because most normal anatomic structure signals are suppressed [9]. Its role as modality is still controversial. However, it is used as a complementary study [10].

Liquid biopsy from CSF and plasma is a promising minimally invasive approach for genomic cancer profiling by obtaining circulating tumor deoxyribonucleic acid (ctDNA) cells, as opposed to invasive tissue biopsies wherein a single-tumor tissue biopsy is less reliable to obtain the whole mutation spectrum [11]. The molecular configuration in the primary tumor undergoes mutation to cross the BBB via hematogenous spread and cause BMs [6]. This is called the seed and soil hypothesis, and suggests that the presence of ctDNA in plasma predicts its capability to cross the BBB and cause BMs (Figure 1). However, whether this mutated ctDNA arises from BMs into the bloodstream is still questionable [1,12].

## 5. Imaging Guidelines (When and How)

The early detection of BMs leads to earlier interventions (Stereoradio-surgery (SRS)) vs. Whole brain Radiotherapy (WBRT)), resulting in better OS. Thus standard imaging protocol [13] is crucial for upfront therapy decision-making, the assessment of response and toxicity, and appearances of BMs before, during, and after treatment. Although CT is the initial modality of imaging when a patient presents with acute neurological symptoms as it is faster and more available [14], Magnetic Resonance Imaging (MRI) including conventional and advanced imaging sequences, and positron emission tomography (PET) techniques are modalities of choice to diagnose and prognosticate BMs along with anatomical, functional, and vascular information. Table 1 outlines the MRI imaging protocol typically used for imaging of brain metastases [15]. The utility of advanced sequences and PET imaging is delineated in Table 2.

Response assessment using 1.5 mm slice thickness MRI with high magnetic field strength and delayed imaging of 15–20 minutes is recommended [15]. The frequency of imaging with regards to histology and post treatment findings are elaborated in Table 3 [16].

The RANO-BM guidelines for BMs [17] and RECIST 1.1 for solid tumors outside CNS are widely accepted. The new RANO-BM guidelines provide bicompartmental criteria for CNS and extra-CNS responses (Table 4) [18] independently because of their differential response to treatment considering divergent acquired mutational evolutions in BMs. Overall CNS response assessment must include target (quantitative) and non-target (qualitative) lesions [19]. Thus, the treatment should always be individualized with a multidisciplinary approach.

### 5.1. Role of Screening

In extracranial metastatic disease, CNS screening is mandatory if it affects the treatment plan. All baseline assessments for certain histological primary tumors should be performed close to ≤4 weeks before the treatment starts [18,20]. In all patients of lung cancers with curative intent, MRI Brain must be included in the imaging protocol.

### 5.2. Role of Imaging in Diagnosis

Prompt imaging for any new/worsening symptomatology is the rule of thumb. The total volume of BMs is a better predictor than the number [21]. Noting measurable lesions (5–10 mm as one of the diameters) forms the baseline for interval assessment. In previous protocols, the size of the target lesion had to be 10 mm to allow feasible quantitative analysis on response scans [12]. However, with rapid advancement in imaging techniques using thin slice and higher resolution, recent studies use 5 mm for target lesions and have shown promising results [11]. Also, mentioning location in relation to eloquent structures and mass effect is needed in preoperative cases to predict OS post treatment [22].

### 5.3. Role of Imaging in Follow Up

Post-operative MRI is performed <48–72 hours after surgery to assess surgery-related hemorrhage/ischemia and the extent of residual disease seen as an abnormal enhancement of the resection cavity to plan further resurgery/SRS [23]. Only 20% of residual cases that are prone to recur are visible on postoperative MRI. Hence, a strict follow up protocol is recommended because the interpretation of any changes in BMs (size/edema/enhancement) depends on the method and duration of treatment [24]. During treatment, non-target lesions (those non measurable or non-quantifiable) may progress (visually in extent) and in some cases this merits discontinuation of therapy despite stability in target lesions as per RECIST criteria.

## 6. Aspects of Imaging with Various Treatment Methods of BMs

### 6.1. Radiation

SRS is surgical resection of BMs followed by high dose radiation to the surgical bed. WBRT is low dose radiation to the whole brain in non resectable BMs. The importance of volume and size over numbers to plan management is shown in Figure 2 [5] Both the above achieve local control, although SRS shows distant failure due to microscopic lesions. Thus, systemic therapy for <4 weeks is necessitated for distant intracranial control in case of SRS [25]. SRS for BMs can be single or fractionated dose (multiple low dose sittings), the latter preferred in geriatric, radioresistant cancers with a large total volume (up to 15 cc) [26]. SIM-SRS has a very complex advanced technique of MRI planning with increased rate of relapse. However, treated metastatic control is good [27].

WBRT has no limits on the number of BMs and does not require a planning MRI. It causes neurocognitive deterioration and xerostomia, thus hippocampal avoidance WBRT (HA-WBRT) and parotid gland exclusion is modulated on a planning MRI in advanced cases. The hippocampus bears watching on follow up scans is recommended as it is the common site for relapse. Approximately 20% of the radiated BMs show a transient increase in size for 3–6 months, known as pseudoprogression (PsP) [28].

Radionecrosis (RN) seen 1–3 years post SRS is associated with a large field of radiation, tumor histology, and the use of concurrent immunotherapy. The risk of RN is less with a fractionated dose. However, SRS is avoided overall if the total volume is >10 cm.

Post radiation changes are seen as hyperintense edema with feathery enhancement and sieve-like T2 appearances. Other imaging findings of stroke, intracranial hemorrhage, radiation-induced tumors, and SMART syndrome (Stroke-like Migraine Attacks after Radiation Therapy) are associated with transient speckled enhancement [29].

It is very important to identify post radiation changes from disease progression as radiation therapy is contraindicated for RN and the latter scenario requires an urgent change of treatment.

The features of conventional and advanced MRI sequences are described in Table 3. Early post radiation changes include low ADC (cytotoxic edema) followed by high ADC values. Low early MRI perfusion (at 1 week post radiation) followed by increased perfusion due to rebound indicate a poor responder [28]. In unequivocal cases, MRS < Aminoacid-PET distinguishes RN from disease progression with high accuracy. The above biomarkers are post radiation predictors [30]. There are certain pretreatment prognostic predictors such as poorer prognosis in BMs with sparse perilesional edema (more tumor invasion—Figure 3) and diffusion restriction (lower ADC values) [30].

### 6.2. Surgery

The most widely applied intraoperative imaging technique is intraoperative ultrasound to detect the dense tissue of BMs. Intraoperative neuronavigation techniques for the guidance of BM resection via MRI (cortical mapping, preoperative functional MRI), electro-physiological monitoring/stimulation, awake surgery, or fluorescence-guided surgery with 5-aminolevulinic acid (5-ALA) are promising [23]. Surgical resection is considered in 1–3 cystic radioresistant BMs of >3 cm with mass effects in the posterior fossa. Special concern is needed for the resection of the glial pseudocapsule [31] to reduce the recurrence rate to be taken (Figure 3). There are four types of metastatic infiltration:

Type 0, displacing growth without infiltration, seen in renal cancer;

Type 1, cluster/cohort infiltration without contact with the blood vessels with no break out;

Type 2, diffuse infiltration—single cells or mini-spheres infiltrating the brain parenchyma with one area of break out;

Type 3, angio-cooptive infiltration into the adjacent brain parenchyma along preexisting blood vessels (typical for melanoma) seen as diffuse blurred edges [32].

Surgical resection for recurrent tumors is recommended in limited intracranial and controlled extracranial disease.

### 6.3. Immunotherapy (ICI)

ICI is given as a last resort because of independent molecular alterations in BMs and also resistance of drugs across BBB, limiting its efficacy. Its widely used in BMs of melanoma, NSCLC, colorectal cancers [33]. Research is ongoing on breast cancer, solid tumors with various combinations. Inflammatory-Immune response post ICI causes PsP features on imaging as well as new enhancing lesions, more seen with melanoma and does not conclude disease progression. Hyperprogression is a paradoxical acceleration of tumor growth with at least a twofold increase in size on two consecutive images, described with ICI [3]. Neurologic death is defined as rapid severe radiographic findings in the brain and clinical neurological progression of a life-threatening nature, in the absence of systemic disease progression/symptomatology also associated with some ICI. Systemic therapy for active CNS lesions with intrathecal administration of ICI especially for targeted cells can be tried.

#### 6.3.1. Recent Advances [34]

There are many advancements in terms of biomarkers (pre and post treatment), molecular level of therapy, minimally invasive methods of biopsy or treatment—all in the hope of improving the OS of patients with BMs and hence halting the trend of dismal prognosis for BMs.

The most important advances are as follows:

#### 6.3.2. Brain Metastasis Velocity (BMV)

The total number of BMs since the first SRS in fractions of 1 year. It is strongly associated with poor OS and neurologic death. It is a surrogate marker for intracranial control.

Patients with BMV > 13 BMs/year have the worst prognosis [35].

#### 6.3.3. Cellular MRI Using Iron Oxide Nanoparticles

Noninvasive imaging of targeted cells and cellular processes by shortening T2 and T2* relaxation, providing improved contrast at micromolar iron concentrations.

#### 6.3.4. Chemical Exchange Saturation Transfer (CEST) Imaging

An MRI contrast method to assess the tumor microenvironment using saturated mobile proteins and peptides that transfer their magnetization to unbound water forming a spectrum.

BMs in general have higher values than the normal brain [36].

Blood Oxygenation Level-Dependent (BOLD) Imaging:

Assesses the magnetic properties of deoxy-hemoglobin and provides functional information.

#### 6.3.5. MRI-Guided Laser Interstitial Thermal Therapy (LITT)

For those treated previously with SRS, LIIT is a minimally invasive treatment option as one can biopsy in the same setting. If the biopsy is RN, then only LIIT will suffice, otherwise fractionated SRS/systemic therapy is selected [37].

#### 6.3.6. Theranostics

Combines the diagnostic and therapeutic properties of radiolabeled compounds to identify nonresponders upfront to save time and unnecessary CNS toxicity [38].

#### 6.3.7. Artificial Intelligence

Because BMs can grow rapidly, they require frequent imaging for detection. This causes an increase in workload (to detect tiny BMs on thinner slice images, and to differentiate mimickers like small vessels), fatigue, and medicolegal challenges for radiologists.

Thus, an automated or semi-automated deep learning-based computer-aided detection (CAD) acts as a second reader to reduce the reporting time, enhance diagnostic performance, and increase vigilance [39]. Studies have shown automatic detection of BM nodules on MRI by CAD up to 3–4 mm, which is more sensitive than a radiologist [40]. Classic machine learning is based on data augmentation and training by signal modulation and segmentation. These techniques facilitate radiomics and future trials to determine the histology and mutation status of lesions [41].

## 7. Imaging Aspects of Specific Primary Tumor Biology

The status of extra-CNS disease in most cancers for OS depends on the number of extracranial metastases and control of primary tumors. However, for CNS, the number is being replaced by volume.

### 7.1. Lung

Non-small cell lung cancer (NSCLC): OS depends on histology (high tendency with adenocarcinoma), hemorrhagic metastases, and sex. Adenocarcinoma of the lung is highly associated with LMD [42].

BMs are usually seen in the parieto-occipital lobes with positive correlation to EGFR-mutated tumors (detected on DTI and T1C-MRI), carcinoembryonic antigen, size of primary tumor, nodal stage, and presence of bone metastases [43].

NSCLC constitutes 85% of all lung cancer types; small cell lung cancer (SCLC) has the highest risk of BMs.

WBRT has been the mainstay of treatment for SCLC BMs in view of widespread micrometastic intracranial disease.

Early post-treatment assessment of interstitial fluid pressure (IFP) and velocity (IFV) can be used to predict the long-term response of lung cancer brain metastases to radiosurgery, allowing timely treatment amendments [44].

### 7.2. Breast

OS depends on the location (predominantly in the cerebellum) of BMs and on LMD. (Figure 4).

Triple-negative breast cancer metastases tend to be cystic/necrotic with shorter OS compared to HER2-positive tumors due to targeted therapies in the latter [45].

BMs tend to occur late in the disease course, and by that point, usually the breast cancer is already chemotherapy-resistant, exacerbated by poor BBB penetration of therapies.

Luminal cancers show better prognosis due to better pharmacokinetics of tamoxifen across the BBB [36,46].

### 7.3. Others

Melanoma: It has highest risk of developing BMs among all solid tumors, often 10 years after diagnosis. (Figure 5) OS depends on age. On MRS, a Choline/Creatinine ratio of <2.0 excludes the possibility of melanoma [47].

Renal cancer: OS depends on a BM-free interval or BM velocity rate [48].

BMs in melanoma or renal cancers with a prior history of WBRT are usually associated with PsP post-SRS [49].

Certain cancers preferentially metastasize to the posterior fossa, including uterine, prostate, and gastrointestinal primary tumors (usually T2 hypointense mucinous type) [50].

## 8. Spine Imaging

Evaluation of CNS metastasis without spinal imaging, if not in all cases, is incomplete. With the location of the dura outside the BBB, the behavior of these lesions is different and need not correlate with the BMs response Figure 6.

Intramedullary spinal cord metastases (ISCM) are more likely to affect elderly individuals with small cell lung cancer and breast cancer with long latency (after 10 years from primary tumor diagnosis) caused by either hematogenous spread, leptomeningeal dissemination, or direct extension (more common) Figure 7 [51,52].

The first therapeutic option of ISCM is radiotherapy (definite efficacy) even for radioresistant tumors such as renal cell carcinoma and melanoma [53]. SRS is promising for limited, oligometastatic disease since the benefits and risks of surgery need to be fully evaluated as no benefit on OS is proven [54,55]. Chemotherapy has little effect on the treatment of ISCM due to the blood–spinal barrier and is reserved for chemotherapy-sensitive tumors such as small cell lung cancer and hematological neoplasms [56]. The exact effect of intrathecal-immunotherapy is still unclear and needs to be further explored [57].

ISCM is a special entity needing more attention with increasing incidence and still grim prognosis.

## 9. Etiopathogenesis

Dural and epidural compartments are supplied directly by the systemic circulation (i.e., no blood–brain barrier and no blood–CSF barrier). Other non-parenchymal areas to metastasize include the calvarium, diploic space, choroid plexus, and pituitary stalk with the gland [58].

EGFR mutated NSCLC LMD shows a complete radiologic response. However, other factors influencing OS must be borne in mind and continued follow up must persist [59].

In breast cancers, LMD relapse is high despite WBRT due to mutational alterations making them chemo/radio resistant.

## 10. Presentation

Metastases to the epidural and leptomeningeal spaces can lead to radicular and cranial nerve palsies, increased intracranial pressure, and back pain.

A sudden onset of weakness and upper motor neuron signs localizing to the spine with or without urinary retention are an emergency.

## 11. Imaging Findings

Leptomeningeal dissemination/seeding is a post craniotomy complication, especially in patients with posterior fossa BMs undergoing a “piecemeal” resection (13.8%) or post SRS [60]. There is no hydrocephalus, and radiation therapy usually suffices.

Classical leptomeningeal disease is an ongoing process of disease progression along the CSF and dura that can need CSF diversion due to hydrocephalus. The disease is seen as linear or nodular or sheet-like subarachnoid deposits along cranial nerves, cerebellar folia, supratentorial sulci, and/or ventricular surfaces, most commonly along the CSF around the basilar artery [61].

Dural-based metastases commonly result from local invasion by skull metastasis, particularly in breast, lung, and prostate cancers, and lymphoma Figure 8 and Figure 9 [62].

Thus, radiographic delineation of the above two entities is critical because WBRT with or without intrathecal chemotherapy is the line of management for LMD.

## 12. Work-Up

Patients undergoing treatment for active CNS metastases are generally followed up with a neurologic examination, MRI of the brain and/or spine every 6–10 weeks, as well as CSF sampling (if LMD is present) [42,63].

The sensitivity of CSF sampling using a lumbar puncture for diagnosis of LMD is 50–60%, with additional sampling of up to 80% [64].

## 13. Management

Treatment is decided according to the neurologic, oncologic, mechanical, and systemic (NOMS) decision framework of the scoring system [65] that states that epidural metastases have better OS and ISCM have the worst OS [66]. With the advantage of no BBB, intrathecal chemotherapy has become more promising. However, the toxicities associated with chemotherapeutic agents should be taken into consideration.

WBRT is associated with local bone changes of myelosuppression.

## 14. Conclusions

CNS metastases have a huge impact on the management of a cancer patient from detection to response assessment apart from prognostication

In this era of molecular and genetic advances, the myth of guarded prognosis on the detection of brain metastasis is lightened, given the evolution of targeted surgeries and therapies with good patient outcomes.

Multimodality imaging is recommended in certain cases because the interpretation of findings depending on primary histology, treatment received, and criteria applied is crucial to further management.

MRI is the investigation of choice in CNS metastasis for screening, diagnosis, and follow up.

Recent advancements in quantitative assessment using MRI, PET, and AI is showing promising results, keeping pace with evolving therapeutic options.

## Figures and Tables

**Figure 1 cancers-16-02667-f001:**
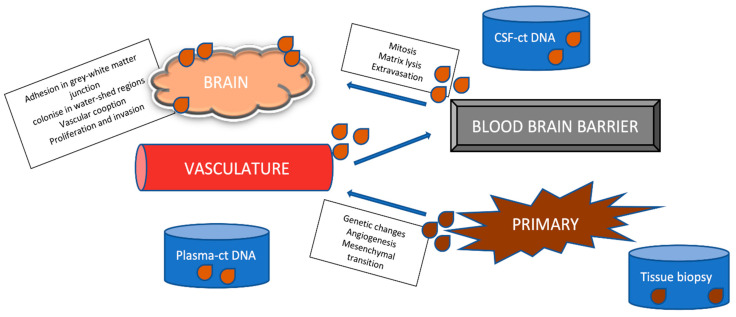
“Seed and soil” hypothesis—etiopathogenesis of BMS.

**Figure 2 cancers-16-02667-f002:**
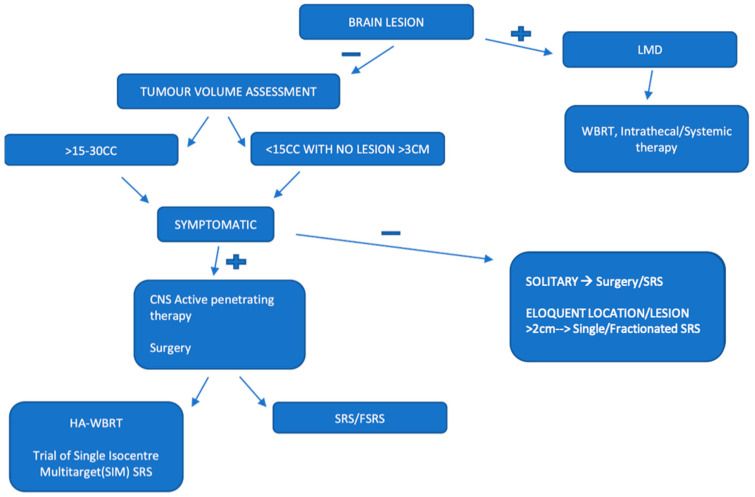
Imaging descriptors under radiation spectrum of BMs.

**Figure 3 cancers-16-02667-f003:**
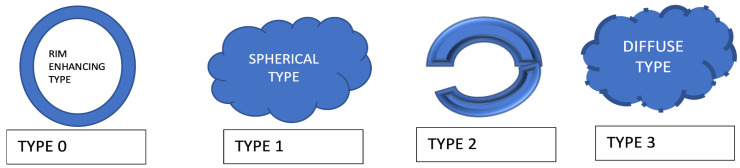
Types of infiltration by brain metastasis.

**Figure 4 cancers-16-02667-f004:**
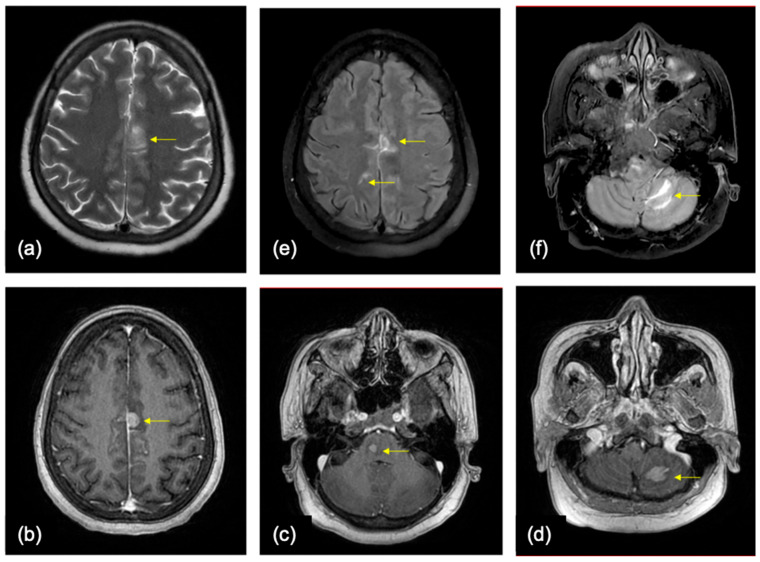
A 50-year-old female—a known case of breast cancer, presented with headache, giddiness, and an episode of seizure. In axial MR images of the brain, (**a**) T2WI shows a dura-based intermediate signal intensity lesion (yellow arrows) along the falx on the left with mild perilesional edema. (**b**–**d**) Axial post contrast T1 FSPGR sequence showing multiple enhancing lesions based on the dura (**b**) and in the parenchyma (**c**) in the pons, and (**d**) in the left cerebellar hemisphere. (**e**,**f**) Axial FLAIR contrast enhanced images showing leptomeningeal disease along the cerebral sulcal spaces and along the cerebellar folia.

**Figure 5 cancers-16-02667-f005:**
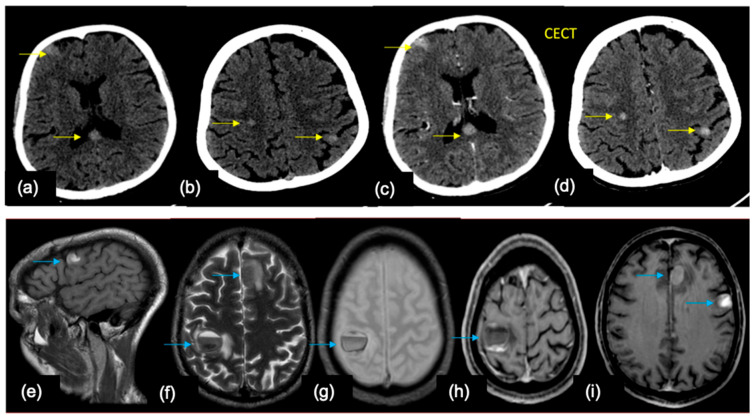
Imaging of two patients with metastatic malignant melanoma Patient 1: NCCT brain in yellow arrows (**a**,**b**) shows multiple hyperdense lesions in the supratentorial parenchyma in both hemispheres in the cortex–subcortical white matter junction with CECT (**c**,**d**) showing enhancement in these lesions. Patient 2: MRI brain from another patient in blue arrows shows (**e**) T1 hyperintense, (**f**) T2 hypointense lesions in similar distribution as Patient 1, with the larger lesion in the right perirolandic region showing the fluid level (suggestive of hemorrhage). (**g**) The finding is confirmed in GRE which shows blooming (**h**,**i**) axial post contrast T1 shows variable enhancement in the lesions.

**Figure 6 cancers-16-02667-f006:**
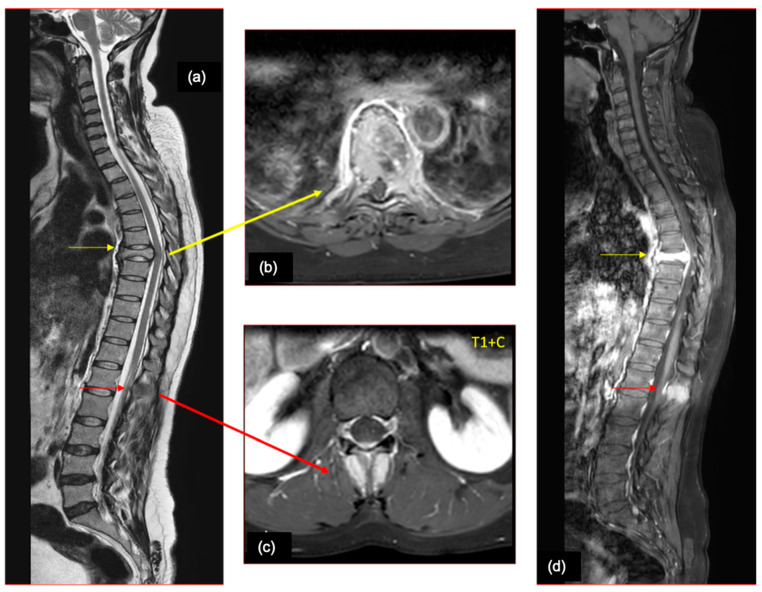
A 52-year-old male—a case of lung cancer, who presented with backache. (**a**) Sagittal T2WI and axial (**b**,**c**) and sagittal (**d**) post contrast T1 fat sat images of the spine show a collapse of the D7 vertebra with retropulsion of the posterior part of the vertebral body causing compression of the cord. Enhancement is noted in the vertebra and epidural space (yellow arrows). Another similar focus is noted in the spinous process of the L1 vertebra (red arrows).

**Figure 7 cancers-16-02667-f007:**
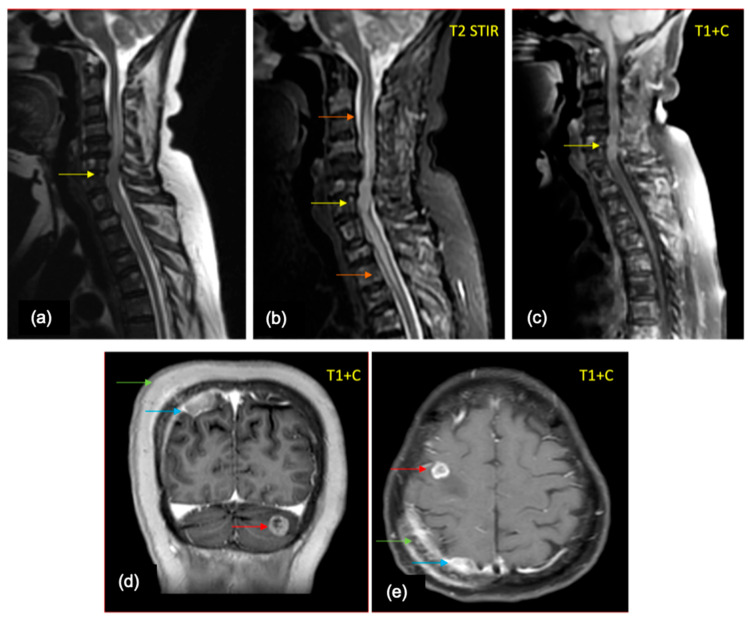
A 64-year-old female—a case of breast cancer, who presented with backache radiating to upper and lower limbs, perioral numbness, and slurred speech (**a**) Sagittal T2WI of the cervical spine shows a T2 intermediate intramedullary lesion (yellow arrow) at the C5-C7 levels. (**b**) Sagittal STIR image shows the lesion with cord edema (orange arrows) extending from the C2 to D5 level. (**c**) Sagittal post contrast T1 fat sat image shows intense post contrast enhancement with multiple enhancing lesions in the visualized vertebral marrow, suggestive of bony metastases. (**d**,**e**) Post contrast T1 coronal and axial images of the brain show calvarial (green arrow), dural (blue arrow), and parenchymal (red arrow) metastases.

**Figure 8 cancers-16-02667-f008:**
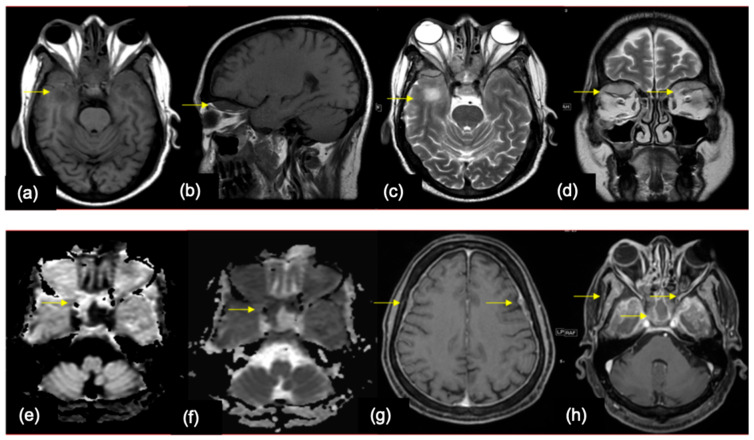
A 59-year-old male—a known case of prostate cancer, presented with headache, facial puffiness, and blurring of vision (**a**) Axial and (**b**) Sagittal T1WI show isointense plaque/sheet like dural thickening involving bilateral temporal convexities with orbital extraconal extension (yellow arrows). (**c**) Axial and (**d**) coronal T2WI show the lesions (yellow arrows) showing intermediate to hyperintense signal with vasogenic edema in underlying parenchyma. (**e**,**f**) DWI and ADC images showing diffusion restriction in the lesions (yellow arrows). (**g**,**h**) Axial T1WI shows post contrast enhancement (yellow arrows).

**Figure 9 cancers-16-02667-f009:**
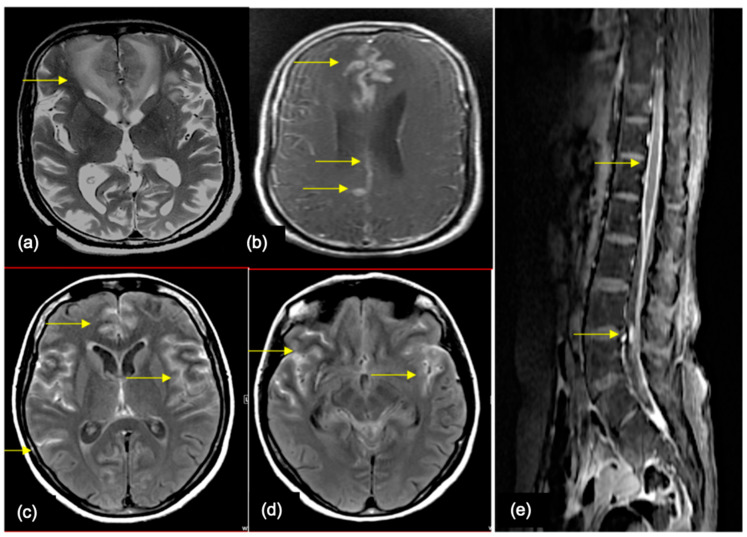
A 45-year-old female—a case of malignant melanoma of the rectum, who presented with headache, imbalance while walking, and vomiting. (**a**) Axial T2WI showing hypointense nodular thickening in the leptomeninges along the anterior interhemispheric fissure with significant surrounding parenchymal vasogenic edema (yellow arrow). (**b**) Axial T1W post contrast image shows intense nodular leptomeningeal enhancement along the interhemispheric fissure (yellow arrow). (**c**,**d**) Axial FLAIR images show diffuse supratentorial leptomeningeal enhancement along the sulcal spaces and basal cisterns (yellow arrows). (**e**) Sagittal post contrast T1 fat sat image of spine showing similar involvement of spinal meninges (yellow arrows)—findings consistent with diffuse leptomeningeal carcinomatosis.

**Table 1 cancers-16-02667-t001:** Brain metastasis protocol.

Sequences	TE (ms)		TR (ms)		FOV (mm)		Slice Thickness (mm)		Technique	
Tesla	1.5T	3T	1.5T	3T	1.5T	3T	1.5T	3T	1.5T	3T
T1WI	Min		2100	550–750	256		≤1.5	1	IR-GRE	TSE
T2WI	80–120		>3500	>2500	240		≤4	3	TSE	
FLAIR	100–140		>6000		240		≤4	3	TSE	
DWI	Min		>5000		240		≤4	3	SS-EPI	
3DT1W-TSE-C	Min		2100	550–750	256		≤1.5	1	IR-GRE	TSE
DSC (optional)		25–35		1000–1500		240		3		GE-EPI

**Table 2 cancers-16-02667-t002:** Radiological imaging modalities and nuclear medicine and advancements.

Modality/Sequence		Indication	Speciality	Disadvantage
Contrast Enhanced Computed Tomography		Excludes neurosurgical emergencies (mass effect/bleed/hydrocephalus)	Bone detail, calcifications, bleed With MRI, assists precise positioning	Radiation, low sensitivity compared to MRI to detect BMs
Magnetic Resonance Imaging Contrast Enhanced		Screening, diagnostic, treating and monitoring.	Gold standard to detect, differentiate and interpret lesions	Overlapping features of post treatment changes with disease
**Magnetic Resonance Imaging—Conventional**				
T1 Weighted Imaging		Assess hemorrhage and enhancement post contrast	Fat suppressed sequence provide calvarial details	Ghosting artefacts, especially in the posterior fossa from the dural venous sinuses, thicker image slices.
T2 Weighted Imaging		Distinguish solid/cystic/necrotic lesions	Basis of many advanced sequences as listed below	Extent underestimated due to bright CSF signal
T2-FLAIR		Perilesional oedema limited to white matter	Detect meningeal involvement in postcontrast FLAIR	Pulsation artefacts
Thin slice spoiled gradient-recalled echo (SPGR) postcontrast MRI		Performed in a head frame for gamma knife treatment planning	Sensitive for the detection of small metastases	Small blood vessels are false positive
**Magnetic Resonance Imaging—Advanced**				
Technique	Biomarker	Correlation	Mechanism	Merit/Demerit(M/D)
Dynamic susceptibility contrast (DSC) MRI	Relative cerebral blood volume (rCBV < 1)	Tumor neoangiogenesis	Changes in T2 or T2* relaxivity acquired rapidly < 2 min	M: acquired in <2 min D: susceptibility artefact from blood products/air/bone or implanted devices in post-op setting
Dynamic contrast enhancement (DCE)MRI	Time curves and Ktrans	Vessel perfusion, permeability, vascular and extravascular volume fractions	Changes in T1 relaxivity	M: assesses antiangiogenic effects of drugs.
Diffusion Weighted Imaging (DWI)	Apparent diffusion coefficient (ADC) maps	Cellularity Cytotoxic and vasogenic oedema	Measuring the displacement of water molecules across the tissue per time unit	M: evaluates post-surgical prognosis, BM growth rate, tumor border zone and differentiates disease from radionecrosis.
Diffusion Tensor Imaging (DTI):	Anisotropic and diffusivity values	White matter tracts	Directed motion of water molecules	M: plans the route of resection in case of eloquent localizations
Magnetic Resonance Spectroscopy (MRS):	Relative and absolute metabolite concentrations (ppm)	Functional interpretation of abnormalities	chemical composition of tissues within the brain environment	M: single-voxel MRS of the peritumoral T2 hyperintense non enhancing area(choline/creatinine ≤ 1.24) Assess response
Arterial spin labelling (ASL)	Cerebral blood flow(CBF)	Blood flow assessed using magnetically labelled blood water protons	No need for exogenous contrast	M: not sensitive to susceptibility artefacts and determine BM recurrence post SRS D: lower signal-to-noise ratio and spatial resolution
**PET Tracer**		**Use**	**Demerit**	
2-[^18^F]-fluoro-2-deoxy-D-glucose (FDG) PET		To distinguish BM relapse from 6 weeks post radiation induced changes	High physiologic glucose metabolism in the normal brain limits diagnostic performance	
Amino Acid PET		Diagnostic performance is superior to both FDG- PET and perfusion–diffusion MRI	Uptake seen in neoplasm and not in normal brain. Crosses blood–brain barrier and differentiates tumor progression from treatment-related changes	
L-[methyl-^11^C]-methionine ([^11^C]MET) PET		Correlates with protein synthesis Uptake is higher in progressive/recurrent BMs than in radionecrosis.	Short half-life of 20 min necessitates an onsite cyclotron for [^11^C]MET production	
O-(2-[^18^F]-fluoroethyl-L-tyrosine ([^18^F]FET) PET		Specific in differentiating tumor from inflammation as its not metabolized into proteins Detects true tumor volume		
L-3,4-dihydroxy-6-[^18^F]fluorophenylalanine ([^18^F] FDOPA) PET		Combined with MRI to distinguish BMs progression or recurrence from radiation necrosis.	Physiological uptake in the corpus striatum prevents clear delineation of BMs in this region.	
[^52^Ga]Ga- dodecane tetra-acetic acid-fibroblast activation protein inhibitor (DOTA-FAPI)		Higher efficacy than ^18^F-FDG PET/CT in detecting BMs	New tracer and limited literature	

**Table 3 cancers-16-02667-t003:** Screening guidelines for BMs in all patients with specific histology and imaging guidelines for BMs—scenario based.

Guidelines For Screening of BMs		Histology	
European Society for Medical Oncology (ESMO)		Non-small-cell lung cancer (NSCLC)	
National Comprehensive Cancer Network (NCCN),		Stage II-IV NSCLC	
British Thoracic Society		Small cell lung cancer of any stage	
National Institute for Health and Care Excellence (NIH)		All lung cancer except Stage 1a NSCLC	
Surveillance, Epidemiology, and End Results (SEER) program		Stage IIIC to IV melanoma	
Joint EANO-ESMO Clinical Practice Guidelines		Metastatic human epidermal growth factor receptor 2 (HER2)-positive and triple-negative breast cancers	
**Scenario Of Treatment**	**Scenario Of Disease Status**	**MRI Interval**	**Remarks**
With SRS	Active intracranial disease	Every 2–3 months for 1 year	Dep on pt/disease factors can increase interval to 4–6 months.
Upfront systemic therapy	Active intracranial disease	Every 6 wks for 3 months → every 9 weeks till 1 year → every 3 months	Target lesions may coalesce or new lesions may form
Immunotherapy	controlled disease	Every 3 months for first year	iRANO criteria is followed
Radionecrosis	Asymptomatic	Repeat MRI in 4–8 weeks with advanced MRI techniques or Amino Acid PET	Focal abnormal signal intensity may never completely resolve.

**Table 4 cancers-16-02667-t004:** New Rano-BM bicompartmental scoring.

Sites of Progression Free Survival	Local Treatment	Loco-Regional Treatment	Systemic Treatment
CNS-l (local CNS)	+	−	−
NON CNS (extracranial)	−	−	+
CNS(local and distant cns)	+	+	+/−
Bi-compartmental (CNS AND NON CNS)	+	+	+

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
