# Peer review of "Imaging Recommendations for Diagnosis, Staging, and Management of Central Nervous System Neoplasms in Adults: CNS Metastases"

_cancers, 2024, doi:10.3390/cancers16152667_

Round 1

Reviewer 1 Report

Comments and Suggestions for Authors

1.Missing abstract;

2.I'm sorry but this is not a goal! In research we are looking for something, we are not familiarizing the readers with something!

"Here, we intend to acquaint the reader with imaging recommendations for CNS secondaries;

3.Please write some conclusions, not "Summary"! What is new in your article must be found in the title, in the conclusions of the abstract and in the final conclusions of the article!

Comments on the Quality of English Language

Minor editing of English language required

Author Response

The changes and responses have been provided in the attached word file along with notes beside the text in main file. 

Reviewer 2 Report

Comments and Suggestions for Authors

The authors provided a  paper about "Imaging recommendations for diagnosis, staging, and management of central nervous system neoplasms in adults: Part II: CNS metastases".

The topic is absolutely relvant especially from a clinical point of view but I have some points which I would like the authors to address a few points as it follows:

1) The format of the article is unsual since there no typical paragraphs (introduction, materials and methods, results, conclusion): in particular I have some doubts about the two paragraphs "4. Imaging referral guidelines" and "6. Imaging guidelines".

2) There are some formatting problems (such as in the case of page 3 lines 81-86) and typos (page 16 line 383 the full stop is missing): please carefully read the manuscript again

3) Figure 3 seems to have some fromatting problem, please provide a higher quality gifure

4) There is no abstract, please provide one

5) There is a huge number of tables but I have the feeling that not all of them are necessary, please consider removing the ones you believe are unnecessary

Author Response

The response to the reviewers comments have been added in the word file provided below and answers to comments have been provided in the notes in the text file.

Round 2

Reviewer 1 Report

Comments and Suggestions for Authors

Nice work! Congratulations!

Reviewer 2 Report

Comments and Suggestions for Authors

The authors shave satisfactorily addressed my previous comments, I have no further suggestions